# UNIFORM: A Unified Deep Learning Framework for Multi-organ and Multi-contrast MRI Reconstruction

**George Yiasemis** [1,2] (iD)                          G.YIASEMIS@NKI.NL
**Jonatan Ferm** [1]                                    J.FERM@NKI.NL
**Nikita Moriakov** [1,2]                               N.MORIAKOV@NKI.NL
**Ritse M. Mann** [1,3]                                 R.MANN@RADBOUDUMC.NL
**Jan-Jakob Sonke** [1,2]                               J.SONKE@NKI.NL
**Jonas Teuwen** [1,2,3]                                J.TEUWEN@NKI.NL

[1] *Netherlands Cancer Institute* [2] *University of Amsterdam* [3] *Radboud University Medical Center*

**Editors:** Accepted for publication at MIDL 2025

## Abstract

MRI is an essential medical imaging modality, yet long acquisition times and organ-specific reconstruction methods often hinder clinical efficiency. In this paper, we propose training a unified deep learning framework (UNIFORM) for reconstructing undersampled multi-coil MRI data across diverse anatomical sites and multiple contrasts. Leveraging a state-of-the-art MRI reconstruction algorithm (vSHARP), UNIFORM was trained on diverse multi-coil $k$-space datasets, including knee, brain, prostate, and cardiac MRI. Evaluated across multiple acceleration factors $(2\times, 4\times, 6\times, 8\times)$, it demonstrated robust performance in terms of quantitative evaluation. Additionally, UNIFORM supports zero-shot self-supervised learning (SSL), enabling effective reconstruction of unseen organs. Zero-shot SSL experiments were conducted on prospectively undersampled breast MRI acquisitions at high acceleration factors $(10\times, 17\times)$, demonstrating improved anatomical detail and reduced noise compared to conventional zero-filling approaches. UNIFORM offers a promising avenue for clinically robust, accelerated multi-organ and multimodal MRI workflows.

**Keywords:** Accelerated MRI Reconstruction, Deep Learning, Multimodal MRI, Zero-shot Self-supervised Learning

## 1. Introduction

MRI's exceptional soft-tissue contrast and non-invasive nature have solidified its role in modern diagnostics, but its slow acquisition process often results in long scan times, patient discomfort, and motion-induced artifacts (Lustig et al., 2008; Zaitsev et al., 2015). While accelerated MRI techniques based on undersampling have been introduced (Pruessmann et al., 1999), traditional model-based methods like compressed sensing and parallel imaging depend on handcrafted priors and are computationally demanding (Lustig et al., 2008; Uecker et al., 2013; Griswold et al., 2002). In contrast, deep learning (DL) techniques have emerged as powerful alternatives, delivering rapid inference and high-fidelity reconstructions (Hammernik et al., 2018; Sriram et al., 2020; Yiasemis et al., 2022). Yet, these methods are typically tailored to specific organs, contrasts, or sampling schemes and accelerations, thereby restricting their scalability and clinical generalization (Huang et al., 2022). To address these challenges, we propose UNIFORM—a unified framework that leverages a single DL model to reconstruct undersampled MRI data across diverse organs and contrasts.

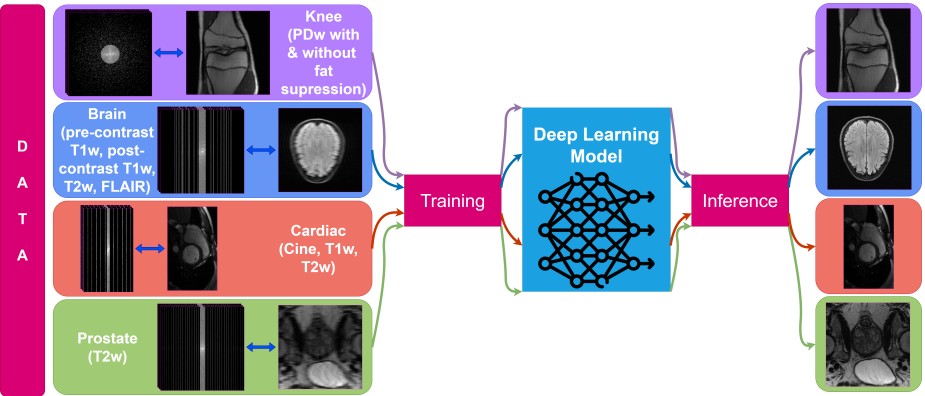

Figure 1: The UNIFORM training and inference pipeline.

## 2. Methods

### 2.1. The UNIFORM Framework

UNIFORM is designed to integrate data from multiple anatomical regions and imaging contrasts into one comprehensive training paradigm. Let $\{\mathcal{D}^j\}_{j=1}^J$, denote the collection of training datasets, each representing a different imaging condition. The optimization objective is expressed as:

$$\boldsymbol{\theta}^* = \underset{\boldsymbol{\theta}}{\operatorname{argmin}} \sum_{j=1}^{J} \sum_{i=1}^{N_j} \mathcal{L}(f_{\boldsymbol{\theta}}(\tilde{\mathbf{x}}_i^j), \mathbf{x}_i^j), \tag{1}$$

where $\mathbf{x}_i^j$ is the fully-sampled ground truth, $\tilde{\mathbf{x}}_i^j$ the undersampled input, and $f_{\boldsymbol{\theta}}$ the parametrized reconstruction model. This formulation ensures that during inference, the model generalizes across diverse distributions. The UNIFORM training and inference pipeline is depicted in Fig. 1. For our experiments, we adopt the 2D variant of the vSHARP algorithm (Yiasemis et al., 2025)—a physics-driven method that fuses variable splitting with iterative ADMM optimization—to efficiently reconstruct multi-coil $k$-space data, with state-of-the-art performance (Lyu et al., 2025; Wang et al., 2025).

**Undersampling**   During training, we retrospectively apply undersampling to fully-sampled $k$-space data. The acceleration factor is randomly selected from $\{2\times, 4\times, 6\times, 8\times\}$, and the undersampling scheme is also chosen randomly to improve generalization. At test time, we evaluate reconstructions at all acceleration factors using fixed undersampling patterns.

**Datasets**   The datasets we employed (Tab. 1) consist of multi-coil $k$-space data from various sources, spanning different anatomical regions and imaging contrasts.

### 2.2. Zero-Shot Self-Supervised Learning on Unseen Data

In scenarios where fully sampled data is impractical, we assess UNIFORM's zero-shot (ZS) generalization on unseen datasets. We employ self-supervised test-time adaptation (Yaman et al., 2023; Yiasemis et al., 2024) using prospectively undersampled data from organs or modalities absent during training. The UNIFORM model is adapted solely with the undersampled data by optimizing a loss function based on the available measurements, eliminating the need for ground truth. In our experiments, we applied ZS-SSL to prospectively undersampled in-house breast T1w data at high acceleration factors ($10\times$ and $17\times$).

| Dataset | Contrasts | Train (#) | Validation (#) | Test (#) |
|---|---|---|---|---|
| fastMRI Knee (Zbontar et al., 2019) | PD with & without fat suppression | 973 | 100 | 99 |
| fastMRI Brain (Zbontar et al., 2019) | T1w, T2w, FLAIR | 4284 | 1577 | 557 |
| fastMRI Prostate (Tibrewala et al., 2023) | T2w | 218 | 48 | 46 |
| CMRxRecon Cardiac (Lyu et al., 2025) | Cine, T1w, T2w | 203 | 229 | 373 |

Table 1: Overview of the training datasets used in UNIFORM.

## 2.3. Training Details

We trained the vSHARP model with default parameters (Yiasemis et al., 2025) over 420k iterations (until validation metrics converged) on two NVIDIA A100 GPUs. The Adam optimizer was employed with default settings, and random augmentations (cropping, flips, rotations) were applied to improve generalizability.

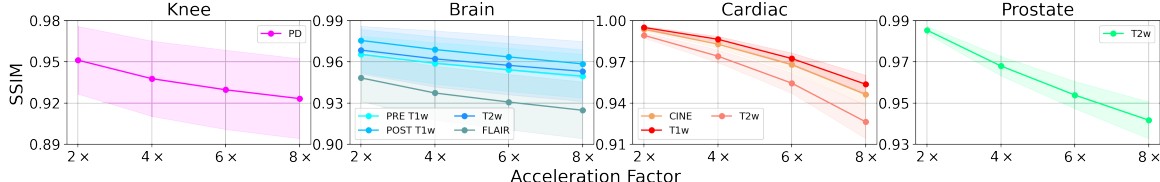

Figure 2: Quantitative performance of UNIFORM on test datasets.

## 3. Results

Reconstruction fidelity was evaluated using SSIM. As shown in Fig. 2, UNIFORM maintained high image quality across different anatomical regions and acceleration factors. Furthermore, the ZS-SSL experiments (Fig. 3) demonstrated that UNIFORM produced reconstructions with superior anatomical detail and reduced noise compared to conventional zero-filling methods.

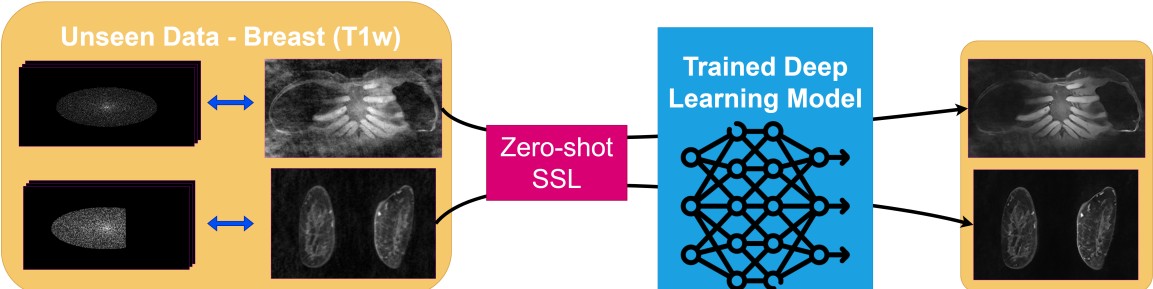

Figure 3: ZS-SSL evaluation on prospectively undersampled breast MRI. **Left:** Undersampled *k*-space & images. **Right:** UNIFORM reconstructions. **Top:** 10×. **Bottom:** 17×.

## 4. Discussion and Conclusion

UNIFORM eliminates the need for organ-specific models by integrating diverse datasets across various organs, contrasts, and acceleration schemes. The zero-shot SSL results further demonstrate the potential of self-supervised learning to adapt UNIFORM to new imaging domains without additional supervised training, effectively leveraging information from datasets used during training. Future work should explore explicitly conditioning the model on input data characteristics, incorporating a wider range of training data, and refining zero-shot self-supervised learning strategies to enhance domain adaptation.

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
