# OpenReview forum: "UNIFORM: A Unified Deep Learning Framework for Multi-organ and Multi-contrast MRI Reconstruction"
_MIDL.io/2025/Short_Papers — MIDL 2025 - Short Papers_

### Official Review · Reviewer_gWDd · 2025-04-29

**Rating:** 4
**Confidence:** 5

**Summary:**

This paper applies vSHARP, a state-of-the-art deep unrolled k-space reconstruction architecture, to a collection of public k-space datasets from multiple organs. The trained network is then adapted to unseen organs using zero-shot adaptation, without access to fully sampled k-space data. Superior image quality is achieved compared to zero-filling methods.

**Strengths:**

•	Demonstrates the effectiveness of training a k-space reconstruction network on a diverse multi-organ dataset, which benefits zero-shot adaptation to unseen organs.

**Weaknesses:**

1.	Lack of novelty: the method simply applies vSHARP to a collection of k-space datasets followed by adaptation.
2.	Lack of comparison to self-supervised learning (SSL) methods without pre-training and to SSL using limited training datasets as part of an ablation study.

---

### Decision · Program_Chairs · 2025-05-01

Accept